# The Economic Nexus between Energy, Water Consumption, and Food Production in the Kingdom of Saudi Arabia

Yosef Alamri [1][ID], Khalid Alrwis [1], Adel Ghanem [1], Sahar Kamara [2], Sharafeldin Alaagib [1,*][ID] and Nageeb Aldawdahi [3]

[1]  Unit of Food Security, Agricultural Economics Department, College of Food and Agricultural Sciences, King Saud University, Riyadh 11451, Saudi Arabia; yosef@ksu.edu.sa (Y.A.); knahar@ksu.edu.sa (K.A.); aghanem@ksu.edu.sa (A.G.)
[2]  Egypt Ministry of Agriculture and Land Reclamation, Agricultural Economics Research Institute, Giza 3751310, Egypt; saharkamara@yahoo.com
[3]  Department of Agricultural Extension and Rural Society, King Saud University, Riyadh 11451, Saudi Arabia; naldawdahi@ksu.edu.sa
*  Correspondence: salaagib@ksu.edu.sa

**Abstract:** The goal of this study was to look at the economic relationship between energy, water use, and plant and animal food production in Saudi Arabia from 1995 to 2020. The results showed that about 55.5%, 82.4%, and 2.5% of changes in the index of plant and animal food production were related to changes in the consumption of water, electricity, and diesel, respectively, using an econometric analysis and the partial correlation coefficient of the second order. The proposed model demonstrated that a 10% change in predicted water, power, or fuel consumption resulted in a 1.97%, 2.78%, and 0.73% change in the index of plant and animal food production, respectively. In light of the Green Middle East Initiative, which intended to minimize carbon emissions, and Saudi agriculture's goal of rationalizing water use, the country's total consumption does not exceed 8 billion m³ of renewable groundwater. This is intended to reduce the use of fuel and increase the use of electricity in the agricultural sector. This rationalizing water consumption, reducing diesel consumption, and expanding electricity consumption affects the production of plant and animal food. In light of the strong interdependence between water, energy, and food production, the agricultural policy has become necessary to increase the amount supplied or available for water to be used in food production, in addition to expanding the production of clean energy and its use in the agricultural sector.

**Keywords:** water; energy; plant and animal food; simple and partial correlation; Saudi agriculture





## 1. Introduction

The agricultural sector contributes to food security and self-sufficiency by playing a vital role in food production. Water resources are some of the most important factors of agricultural productivity, according to one study (Ghanem and Al-Nashwan 2021b), which found that they contributed roughly 23.6% of the entire value of agricultural output from 1990 to 2019. The increasing demand for water for domestic, industrial, and agricultural needs is putting a strain on renewable surface and groundwater resources. The study by Alrwis et al. (2021) was concerned with measuring the impact of the scarcity of water resources on agricultural economic development in the Kingdom of Saudi Arabia. This research found that if there is a scarcity of water resources accessible to the agricultural sector, the overall cultivated area will decrease, lowering the value of agricultural output and GDP. According to the National Water Strategy, if current water consumption trends continue, the water reserve in some regions of the sedimentary shelf may be depleted within the next 12 years (Ministry of Environment, Water and Agriculture 2018).

In addition, energy is needed in agriculture. According to a report by the National Center for Energy Research (2015), Jordan's agricultural sector utilizes roughly 3.2% of the country's energy. Diesel accounted for 5.1% of total consumption in the agricultural sector, gasoline accounted for 0.18%, liquefied gas accounted for 2.4%, and electric power accounted for 1.6%. In the Kingdom of Saudi Arabia, public power is utilized for irrigation in roughly 44.7% of the land. The overall volume of petroleum products utilized in agricultural holdings was around 1.862 million liters, where diesel accounted for approximately 98.0%, gasoline for 0.9%, and oil for 1.0% (General Authority for Statistics 2015). Some farmers have resorted to contracting with the Saudi Electricity Company to light farms and operate wells because of the rise in diesel prices and the implementation of the environmental protection program against pollution. Agricultural subscribers climbed from 54.55 thousand in 2005, accounting for 1.1% of total electricity subscribers (4.96 million), to 97.08 thousand in 2019, accounting for 0.99% of total electricity subscribers (9.76 million) (Saudi Central Bank 2021).

Water and energy usage are inextricably tied to achieving food sovereignty. According to a study by Ghanem and Al-Nashwan (2021b), increasing the area planted with palm trees by 10% results in a 9.5% increase in the amount of water utilized to attain food sovereignty for dates. According to a study by Ghanem and Al-Nashwan (2021a), the total amount of water used in grain production was 136.32 billion $m^3$, accounting for 27.0% of total water utilized in the agricultural sector from 1990 to 2020. The study by El-Gafy (2017) examined the relationship between water, food, and energy by using several indicators that consider water and energy consumption, total productivity, and economic productivity. This study showed that the water-food-energy nexus index (WFENI) for summer crops in Egypt ranged from a minimum of 0.21 for rice to a maximum of 0.79 for onions.

A review of the findings of past studies discovered that research and economic studies on the link between water and energy usage on one hand and food production on the other are sparse and inaccessible. As a result, the focus of this research was on determining the economic interdependence of water and energy in the production of plant and animal food in Saudi Arabia.

The Food and Agriculture Organization (2014) indicated that water, energy, and food are essential elements for human well-being, poverty reduction, and sustainable development. In light of ongoing population growth, the demand for water, energy, and food will increase over the coming decades. Global energy consumption is expected to increase by 50% by 2035, and water consumption for agricultural purposes is expected to increase by 10% by 2050.

Claudia Ringler et al. (2016) addressed the General Assembly of the United Nations on the water-energy-food (WEF) nexus. Their suggested goals and related targets for 2030 included (1) end hunger, achieve food security and improved nutrition, and promote sustainable agriculture (SDG2); (2) ensure the availability and sustainable management of water and sanitation for all (SDG6); and (3) ensure access to affordable, reliable, sustainable, and modern energy for all (SDG7). There will be tradeoffs between achieving these goals particularly in the wake of changing consumption patterns and rising demands from a growing population expected to reach more than nine billion by 2050. This paper uses global economic analysis tools to assess the impacts of long-term changes in fossil fuel prices, for example, as a result of a carbon tax under the UNFCCC or in response to new, large findings of fossil energy sources, on water and food outcomes. We find that a fossil fuel tax would not adversely affect food security and could be a boon to global food security if it reduces adverse climate change impacts.

A study by Mahlknecht et al. (2020) showed that achieving sustainable development in Latin America and the Caribbean depends on improving the prices of food commodities, in addition to paying attention to energy, water, and food security. This study also showed an increase in the need to develop infrastructure to reduce energy consumption and to produce clean energy. Water scarcity is expected to increase in light of the instability of

rainfall, which requires improved water management and availability and the promotion of good agricultural practices and sustainable food systems.

Saul Ngarava (2021) studied the relationship between water-energy-food (WEF) nexus and margins for the lateral transmission of price volatilities within several sectors. The problem was that any inflationary price tendencies in one of the WEF sectors will have direct and indirect effects on the others. The objective of the study was to determine the relationships between inflation in food, energy, and water and determine whether there were spillovers in South Africa. Monthly consumer price indices for food, energy, and water for the period from January 2002 to December 2020 were used. The parsimonious vector autoregressive (VAR) model was used in the data analysis. The study found that prior to 2013, the inflation rate was higher for food relative to water and to energy, separately. After 2017, water had a higher inflation rate relative to energy and to food, separately. Furthermore, energy inflation had a positive impact on both water inflation and food inflation, while water inflation also had positive impact on food inflation. The study concludes that there is a nexus in the lateral inflation between food, energy, and water. Its recommendations included building resilience within the nexus by decoupling food and other sectors from fossil-fuel-derived energy.

Ziyu Pan et al. (2021) studied the shortage of water resources that restrict the economic development in Northwest China. Guiding the decoupling between regional economic development and water consumption is a critical way to achieve sustainable development. Based on the analysis of the food and energy production value and their water consumption in Northwest China from 2009 to 2019, this paper used the Tapio model to analyze the decoupling relationship between food, energy production, and water consumption and used factors derived from the logarithmic mean divisional index (LMDI) model that affect decoupling. The results showed that most water consumption for food and energy production in Northwest China was out of the ideal strong decoupling, the decoupling status was unstable, and recoupling occurred frequently. The increase in water intensity and the change in industrial structure were the promoting factors of decoupling between production value and water consumption in food and energy in Northwest China, while the increase in production value and the increase in population size were the main restraining factors. Therefore, in search of strong decoupling, the government should guide the food and energy industry to move toward implementing water-saving measures in policies and promote the enthusiasm and efficiency of the labor force through financial support and other ways. Moreover, ecological protective measures, such as water source protection and sewage treatment, need to be strengthened.

By reviewing the methods and results of previous studies, it was found that some studies relied on the calculation of simple correlation coefficients, while others used the one-equation model. This study can be distinguished from all previous studies in that it used partial correlation coefficients of the first and second order, and it also used a proposed model consisting of four behavioral equations, which include internal and external variables, in order to be more comprehensive in studying the interdependence between water, energy, and food production. It also shows the scarcity and lack of economic studies in the field of the interdependence between water and energy consumption on one hand and food production on the other in the Kingdom of Saudi Arabia. Therefore, this study focused on measuring this economic interdependence.

Research Objectives:

The goal of this study was to look at the economic nexus between agricultural production, water usage, and energy (diesel and electricity) consumption in the Kingdom of Saudi Arabia from 1995 to 2020.

1    The current state of water and energy use and that of plant and animal food production.
2    Calculation of the amount and value of water and energy productivity in Saudi agriculture.

3     Calculation of the first- and second-order simple and partial correlation coefficients between the value of agricultural output and the index for plant and animal food production, as well as water and energy consumption in Saudi agriculture.

4     Estimation of the proposed model for assessing Saudi agriculture's economic connection between food production on one hand and water and energy use on the other.

## 2. Methodology

This research relied on data published in (1) the Saudi Ministry of Environment, Water and Agriculture's statistical book, (2) the Saudi Central Bank's yearly reports, (3) the website of the Food and Agriculture Organization (FAO), and (4) the Saudi Electricity Company's reports. Econometric analysis was also used in this study. The first- and second-order simple and partial correlation coefficients between water, energy, and plant and animal food production were employed as follows (Gujarati and Porter 2009):

After removing the effect of the variable $X_2$, the partial correlation coefficient of the first order between $YX_1$ was determined as follows:

$$r_{YX_1 l X_2} = \frac{r_{YX_1} - r_{YX_2} r_{X_1 X_2}}{\sqrt{\left(1 - r_{YX_2}^2\right)\left(1 - r_{X_1 X_2}^2\right)}}$$

The partial correlation coefficient of the first order between $YX_2$, after excluding the effect of the variable $X_1$, was calculated as follows:

$$r_{YX_2 l X_1} = \frac{r_{YX_2} - r_{YX_1} r_{X_1 X_2}}{\sqrt{\left(1 - r_{YX_1}^2\right)\left(1 - r_{X_1 X_2}^2\right)}}$$

The partial correlation coefficient of the second order between $YX_1$, after excluding the effect of the two variables $X_2$ $X_3$, was calculated as follows (Ismail 2001):

$$r_{YX_1 / X_2 X_3} = \frac{r_{YX_1 / X_2} - r_{YX_3 l X_2} r_{X_1 X_3 / X_2}}{\sqrt{\left(1 - r_{YX_3 / X_2}^2\right)\left(1 - r_{X_1 X_3 / X_2}^2\right)}}$$

The second-order partial correlation coefficient between $YX_2$, after excluding the effect of the two variables $X_1$ $X_3$, was calculated as follows:

$$r_{YX_2 / X_1 X_3} = \frac{r_{YX_2 / X_1} - r_{YX_3 l X_1} r_{X_2 X_3 / X_1}}{\sqrt{\left(1 - r_{YX_3 / X_1}^2\right)\left(1 - r_{X_2 X_3 / X_1}^2\right)}}$$

The partial correlation coefficient of the second order between $YX_3$, after excluding the effect of the two variables $X_1$ $X_2$, was calculated as follows:

$$r_{YX_3 / X_1 X_2} = \frac{r_{YX_3 / X_1} - r_{YX_2 l X_1} r_{X_2 X_3 / X_1}}{\sqrt{\left(1 - r_{YX_2 / X_1}^2\right)\left(1 - r_{X_2 X_3 / X_1}^2\right)}}$$

The proposed model for studying the economic nexus between energy and water consumption on one hand and food production on the other in the Kingdom of Saudi Arabia during the period 1995–2020 was also estimated. The proposed model consists of the following behavioral equations:

$$Y_1 = a_0 + a_1 X_1 + e_1$$
$$Y_2 = b_0 + b_1 X_2 + e_2$$
$$Y_3 = c_0 + c_1 X_3 + e_3$$
$$Y_4 = d_0 + d_1 \hat{Y}_1 + d_2 \hat{Y}_2 + d_3 \hat{Y}_3 + e_4$$

The proposed model includes the following variables:

- Four endogenous variables—the amount of water used for agricultural purposes in billion m³ ($Y_1$), electricity consumption in the agricultural sector in gigawatt-hours ($Y_2$), diesel consumption in the agricultural sector in million barrels ($Y_3$), and the index for the production of plant and animal food ($Y_4$).
- Three exogenous variables—the cropped area ($X_1$), the total number of projects financed by the Agricultural Development Fund ($X_2$), and the number of agricultural machines and equipment ($X_3$). Because the number of machines and engines was unavailable, the value of the fixed capital of machines and engines in billion riyals was utilized as a substitute.

The equations of the proposed model were estimated by using the ordinary least squares (OLS) method, where the diameter of the matrix of internal variables of the proposed model was 1 and all numbers above the diameter were 0 (Gujarati and Porter 2009):

| Equation | Endogenous Variables | | | |
| --- | --- | --- | --- | --- |
| | $Y_1$ | $Y_2$ | $Y_3$ | $Y_4$ |
| First | 1 | 0 | 0 | 0 |
| Second | 0 | 1 | 0 | 0 |
| Third | 0 | 0 | 1 | 0 |
| Fourth | $d_1$ | $d_2$ | $d_3$ | 1 |

## 3. Results and Discussion

*3.1. The Current Situation of Energy and Water Consumption and the Production of Plant and Animal Food*

3.1.1. The Current Status of Energy Consumption in the Agricultural Sector

When looking at the evolution of energy consumption in the agricultural sector, the data in Table 1 show that electricity consumption in the agricultural sector increased from 1602.8 gigawatts-hours in 1995, representing 1.87% of total electricity consumption, to 5150.0 gigawatt-hours in 2020, representing 1.78% of total electricity consumption. The number of machines and combine harvesters used in Saudi agriculture decreased as a result of rising diesel prices and the state's adoption of a strategy to preserve the environment by reducing the area planted with wheat, barley, and green fodder, and thus, diesel consumption decreased from 14.59 thousand barrels in 1995 to 8.65 thousand barrels in 2020.

**Table 1.** The relative importance of energy consumption in the agricultural sector during the period 1995–2020.

| Year | Electricity in Gigawatt-Hours | | | Diesel in Thousand Barrels |
| --- | --- | --- | --- | --- |
| | Agricultural | Total | % | |
| 1995 | 1602.8 | 85,908 | 1.87 | 14.59 |
| 1996 | 1714.7 | 89,641 | 1.91 | 13.14 |
| 1997 | 1708.5 | 92,228 | 1.85 | 14.15 |
| 1998 | 1909.1 | 97,050 | 1.97 | 12.66 |
| 1999 | 2147.9 | 105,612 | 2.03 | 13.74 |
| 2000 | 2265.4 | 114,161 | 1.98 | 12.54 |
| 2001 | 2379.5 | 122,944 | 1.94 | 13.57 |
| 2002 | 2640.0 | 128,629 | 2.05 | 13.71 |
| 2003 | 2666.0 | 142,194 | 1.87 | 13.62 |
| 2004 | 2920.0 | 144,385 | 2.02 | 13.13 |
| 2005 | 3136.3 | 153,283.6 | 2.05 | 12.40 |

**Table 1.** *Cont.*

| Year | Electricity in Gigawatt-Hours | | | Diesel in Thousand Barrels |
|---|---|---|---|---|
| | Agricultural | Total | % | |
| 2006 | 3380.3 | 163,151.1 | 2.07 | 12.03 |
| 2007 | 3373.6 | 169,302.8 | 1.99 | 12.04 |
| 2008 | 3689.3 | 179,272.2 | 2.06 | 10.88 |
| 2009 | 5329.0 | 193,471.3 | 2.75 | 9.35 |
| 2010 | 3760.9 | 212,262.6 | 1.77 | 9.04 |
| 2011 | 3941.9 | 219,661.6 | 1.79 | 8.81 |
| 2012 | 4361.9 | 240,288.1 | 1.82 | 8.35 |
| 2013 | 4290.3 | 256,687.6 | 1.67 | 7.78 |
| 2014 | 4577.4 | 274,502.2 | 1.67 | 11.73 |
| 2015 | 5167.5 | 286,036.8 | 1.81 | 11.63 |
| 2016 | 5380.6 | 287,692.3 | 1.87 | 11.50 |
| 2017 | 5653.3 | 288,656.8 | 1.96 | 10.08 |
| 2018 | 4905.4 | 289,822.2 | 1.69 | 9.74 |
| 2019 | 4984.8 | 279,677.5 | 1.78 | 9.61 |
| 2020 | 5150.0 | 289,328.0 | 1.78 | 8.65 |
| Average | 3578.3 | 188,686.5 | 1.92 | 11.48 |

Source: (1) Saudi Central Bank (2021) annual statistics for 2020, as of 1 June 2021; (2) Saudi Electricity Company (1995–2020) annual reports.

### 3.1.2. The Current Status of Water Consumption for Agricultural Purposes

By studying the development of water consumption in the agricultural sector, it is clear from the data in Table 2 and Figure 1 that despite the decrease in the cropped area from 1302.4 thousand hectares in 1995 to 694.6 thousand hectares in 2013, the water consumption in the agricultural sector increased from 14.82 billion m$^3$ in 1995 to 18.64 billion m$^3$ in 2013, and thus, the average share per hectare increased from 11.38 thousand m$^3$ in 1995 to 26.84 thousand m$^3$ in 2013. This was due to the decrease in the area planted with wheat, in accordance with Resolution 335, and farmers' tendency to expand the cultivation of green fodder, depleting the water. Because of the increase in the cropped area to 1038.12 thousand hectares, the amount of water used amounted to 20.83 billion m$^3$ in 2015, then the cropped area decreased to 771.92 thousand hectares, and then the amount of water used decreased to 8.5 billion m$^3$, at a rate of 11.01 thousand m$^3$/hectare in 2020.

**Table 2.** Crop area, water quantity, and total value of agricultural production in the Kingdom of Saudi Arabia during the period 1995–2020.

| Year | Water Used for Agricultural Purposes in Billion m$^3$ | Crop Area in Thousand Hectares | The Average Share of a Hectare of Water per Thousand m$^3$/Hectare | The Value of Agricultural Production at Constant Prices in Millions of Riyals |
|---|---|---|---|---|
| 1995 | 14.82 | 1302.4 | 11.38 | 38,062 |
| 1996 | 15.32 | 1173.3 | 13.06 | 37,939 |
| 1997 | 18.66 | 1263.3 | 14.77 | 39,091 |
| 1998 | 18.05 | 1130.7 | 15.96 | 39,468 |
| 1999 | 18.30 | 1226.5 | 14.92 | 40,367 |
| 2000 | 18.00 | 1119.9 | 16.07 | 41,945 |
| 2001 | 18.64 | 1211.6 | 15.38 | 42,182 |
| 2002 | 18.28 | 1224.5 | 14.93 | 42,724 |
| 2003 | 18.03 | 1216.0 | 14.83 | 43,072 |

**Table 2.** *Cont.*

| Year | Water Used for Agricultural Purposes in Billion m³ | Crop Area in Thousand Hectares | The Average Share of a Hectare of Water per Thousand m³/Hectare | The Value of Agricultural Production at Constant Prices in Millions of Riyals |
|---|---|---|---|---|
| 2004 | 19.85 | 1172.7 | 16.93 | 44,616 |
| 2005 | 18.59 | 1106.8 | 16.80 | 45,088 |
| 2006 | 17.00 | 1074.2 | 15.83 | 45,544 |
| 2007 | 15.42 | 1075.0 | 14.34 | 46,431 |
| 2008 | 15.08 | 971.6 | 15.52 | 47,048 |
| 2009 | 14.75 | 835.0 | 17.66 | 47,533 |
| 2010 | 14.41 | 806.7 | 17.86 | 52,098 |
| 2011 | 15.97 | 786.8 | 20.30 | 54,565 |
| 2012 | 17.51 | 745.5 | 23.49 | 56,096 |
| 2013 | 18.64 | 694.6 | 26.84 | 57,936 |
| 2014 | 19.61 | 1047.4 | 18.72 | 59,382 |
| 2015 | 20.83 | 1038.12 | 20.07 | 59,744 |
| 2016 | 19.79 | 1026.91 | 19.27 | 60,122 |
| 2017 | 19.20 | 900.06 | 21.33 | 60,422 |
| 2018 | 19.00 | 869.91 | 21.84 | 60,501 |
| 2019 | 10.50 | 857.76 | 12.24 | 61,202 |
| 2020 | 8.50 | 771.92 | 11.01 | 60,187 |
| Average | 17.03 | 1024.97 | 16.98 | 49,360.19 |

Source: (1) Ministry of Environment, Water and Agriculture (2020), statistical book, 1995–2020; (2) Saudi Central Bank, annual statistics 2020, as of 6 January 2021. Constant prices (2010 = 100).

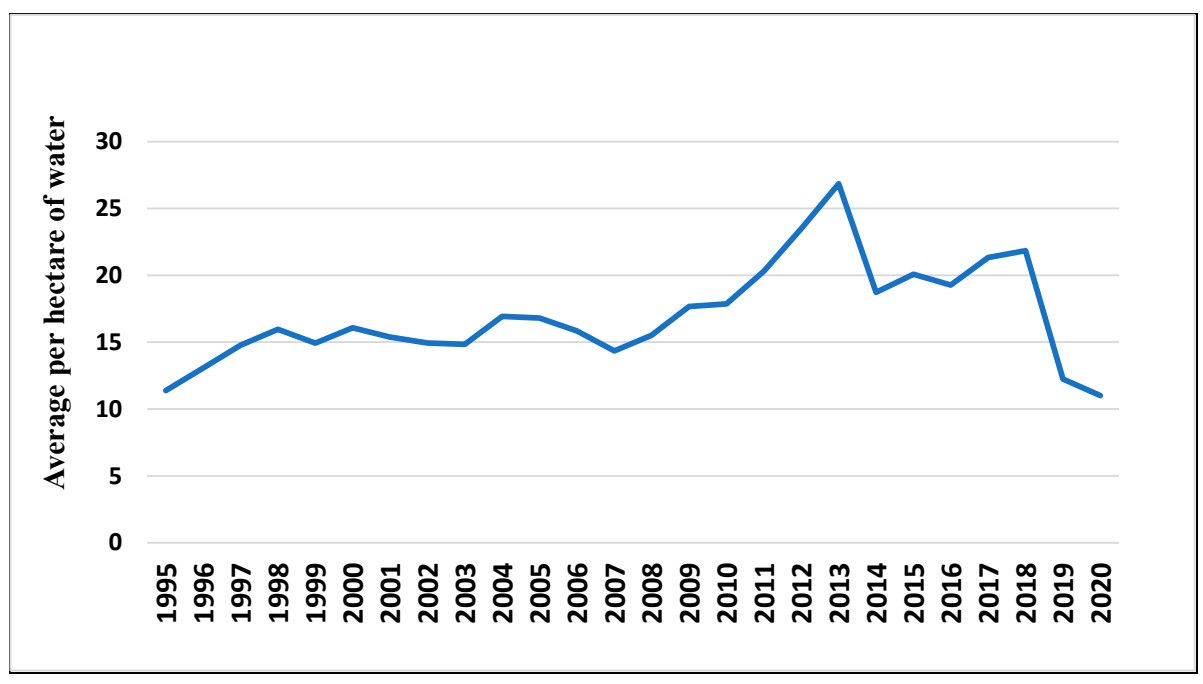

**Figure 1.** The average share of water per hectare per thousand m³ during the period 1995–2020. Source: The data in Table 2.

### 3.1.3. The Current Status of Plant and Animal Food Production

By studying the development of plant food production (cereals, fruits, and vegetables) and animal production (red meat, poultry meat, fish, milk, and eggs) during the period 1995–2020, it is clear from the data in Tables 3 and 4 and Figure 2 that the production of

vegetable food decreased from 6416.4 thousand tons in 1995 to 4102.7 thousand tons in 2019 and then increased to 6289.9 thousand tons in 2020. Calculating the index of vegetable food production, it is clear that vegetable food production decreased in 2019 from its estimated counterpart in 1995, at a rate of 36.1%. The total production of meat (red meat, poultry, and fish) increased from 592 thousand tons in 1995 to 1354 thousand tons in 2020. By calculating the index of the total meat production, it is clear that the total meat production increased in 2020 over its estimated counterpart in 1995, at a rate of 128.7%. Milk production also increased from 698 thousand tons in 1995 to 2911 thousand tons in 2020. According to a calculation of the milk production index, it is clear that milk production in 2020 increased from its estimated counterpart in 1995, at a rate of 317.0%. Egg production also increased from 132 thousand tons in 1995 to 350 thousand tons in 2020. According to a calculation of the egg production index, it is clear that egg production in 2020 increased from its estimated counterpart in 1995, at a rate of 165.2%. Finally, the geometric mean of the index of vegetable food production and the overall output of meat, milk, and eggs was used to determine the general index of food production. Table 4 shows that overall food production (plant and animal) rose by 123.1% in 2020 compared to 1995, implying an annual growth rate of 4.92% from 1995 to 2020.

**Table 3.** Production of plant and animal food for the Kingdom of Saudi Arabia in thousand tons during the period 1995–2020.

| Year | Cereals | Fruits | Vegetables | Vegetarian Food | Red Meat | Poultry Meat | Fish | Total Meat | Milk | Eggs |
|---|---|---|---|---|---|---|---|---|---|---|
| 1995 | 2671 | 1053 | 2693 | 6416.4 | 154 | 390 | 48 | 592 | 698 | 132 |
| 1996 | 1934 | 1092 | 2631 | 5656.6 | 155 | 397 | 51 | 603 | 749 | 125 |
| 1997 | 2341 | 1151 | 2600 | 6091 | 157 | 451 | 54 | 662 | 816 | 131 |
| 1998 | 2205 | 1150 | 2137 | 5491 | 157 | 395 | 55 | 607 | 883 | 136 |
| 1999 | 2488 | 1133 | 1896 | 5516.8 | 159 | 418 | 52 | 629 | 937 | 136 |
| 2000 | 2172 | 1188 | 1927 | 5287.1 | 160 | 483 | 55 | 698 | 1039 | 129 |
| 2001 | 2594 | 1210 | 2107 | 5910.6 | 160 | 521 | 61 | 737 | 1067 | 138 |
| 2002 | 2856 | 1241 | 2137 | 6234.4 | 162 | 467 | 64 | 686 | 1139 | 138 |
| 2003 | 2951 | 1331 | 2214 | 6496.6 | 165 | 468 | 67 | 700 | 1200 | 137 |
| 2004 | 3194 | 1454 | 2479 | 7127 | 167 | 522 | 67 | 756 | 1232 | 145 |
| 2005 | 3004 | 1554 | 2571 | 7129 | 169 | 537 | 75 | 781 | 1338 | 169 |
| 2006 | 3042 | 1549 | 2617 | 7208 | 170 | 535 | 81 | 786 | 1381 | 174 |
| 2007 | 2967 | 1582 | 2596 | 7145 | 171 | 508 | 91 | 770 | 1436 | 188 |
| 2008 | 2438 | 1616 | 2696 | 6750 | 170 | 446 | 93 | 709 | 1690 | 170 |
| 2009 | 1592 | 1619 | 2676 | 5889 | 171 | 494 | 96 | 761 | 1718 | 191 |
| 2010 | 1571 | 1549 | 2521 | 5641 | 172 | 447 | 100 | 719 | 1763 | 219 |
| 2011 | 1418 | 1609 | 2648 | 5675 | 171 | 529 | 76 | 776 | 1838 | 220 |
| 2012 | 1085 | 1639 | 2650 | 5374 | 173 | 588 | 90 | 851 | 1872 | 220 |
| 2013 | 883 | 1688 | 2729 | 5300 | 174 | 604 | 90 | 868 | 1943 | 240 |
| 2014 | 925 | 1089 | 2282 | 4296 | 248 | 507 | 92 | 847 | 2378 | 210 |
| 2015 | 1630 | 1319 | 1847 | 4795.9 | 258 | 518 | 104 | 880 | 2399 | 275 |
| 2016 | 1525 | 1462 | 1925 | 4911.3 | 262 | 554 | 107 | 923 | 2422 | 280 |
| 2017 | 1171 | 1643 | 1480 | 4292.9 | 267 | 540 | 121 | 928 | 2446 | 285 |
| 2018 | 1063 | 1715 | 1440 | 4217.9 | 270 | 554 | 140 | 964 | 2363 | 286 |
| 2019 | 967 | 1738 | 1398 | 4102.7 | 275 | 800 | 142 | 1217 | 2683 | 349 |
| 2020 | 1255 | 2342 | 2695 | 6289.9 | 288 | 900 | 166 | 1354 | 2911 | 350 |
| Average | 1997.8 | 1450.6 | 2292 | 5740.2 | 192.5 | 522 | 86.1 | 800.2 | 1628.5 | 199 |

Source: Food and Agriculture Organization (1995–2020) website.

**Table 4.** The index of plant and animal production for the Kingdom of Saudi Arabia during the period 1995–2020.

| Year | Cereals | Fruits | Vegetables | Plant Production | Animal Production | | | Food Production Index |
|------|---------|--------|------------|------------------|-------------------|------|------|------------------------|
| | | | | | Total Meat | Milk | Eggs | |
| 1995 | 100 | 100 | 100 | 100 | 100 | 100 | 100 | 100 |
| 1996 | 72.4 | 103.7 | 97.7 | 88.2 | 101.9 | 107.3 | 94.7 | 97.7 |
| 1997 | 87.6 | 109.3 | 96.5 | 94.9 | 111.8 | 116.9 | 99.2 | 105.3 |
| 1998 | 82.6 | 109.2 | 79.4 | 85.6 | 102.5 | 126.5 | 103 | 103.4 |
| 1999 | 93.1 | 107.6 | 70.4 | 86 | 106.3 | 134.2 | 103 | 106 |
| 2000 | 81.3 | 112.8 | 71.6 | 82.4 | 117.9 | 148.9 | 97.7 | 109 |
| 2001 | 97.1 | 114.9 | 78.2 | 92.1 | 124.5 | 152.9 | 104.5 | 116.4 |
| 2002 | 106.9 | 117.9 | 79.4 | 97.2 | 115.9 | 163.2 | 104.5 | 117.7 |
| 2003 | 110.5 | 126.4 | 82.2 | 101.2 | 118.2 | 171.9 | 103.8 | 120.9 |
| 2004 | 119.6 | 138.1 | 92.1 | 111.1 | 127.7 | 176.5 | 109.8 | 128.8 |
| 2005 | 112.5 | 147.6 | 95.5 | 111.1 | 131.9 | 191.7 | 128 | 137.7 |
| 2006 | 113.9 | 147.1 | 97.2 | 112.3 | 132.8 | 197.9 | 131.8 | 140.4 |
| 2007 | 111.1 | 150.2 | 96.4 | 111.4 | 130.1 | 205.7 | 142.4 | 143.5 |
| 2008 | 91.3 | 153.5 | 100.1 | 105.2 | 119.8 | 242.1 | 128.8 | 140.8 |
| 2009 | 59.6 | 153.8 | 99.4 | 91.8 | 128.5 | 246.1 | 144.7 | 143.2 |
| 2010 | 58.8 | 147.1 | 93.6 | 87.9 | 121.5 | 252.6 | 165.9 | 145.4 |
| 2011 | 53.1 | 152.8 | 98.3 | 88.4 | 131.1 | 263.3 | 166.7 | 150.2 |
| 2012 | 40.6 | 155.7 | 98.4 | 83.8 | 143.8 | 268.2 | 166.7 | 152.3 |
| 2013 | 33.1 | 160.3 | 101.3 | 82.6 | 146.6 | 278.4 | 181.8 | 157.3 |
| 2014 | 34.6 | 103.4 | 84.7 | 67 | 143.1 | 340.7 | 159.1 | 151 |
| 2015 | 61 | 125.3 | 68.6 | 74.7 | 148.6 | 343.7 | 208.3 | 167.9 |
| 2016 | 57.1 | 138.8 | 71.5 | 76.5 | 155.9 | 347 | 212.1 | 172.2 |
| 2017 | 43.8 | 156 | 55 | 66.9 | 156.8 | 350.4 | 215.9 | 167.8 |
| 2018 | 39.8 | 162.9 | 53.5 | 65.7 | 162.8 | 338.5 | 216.7 | 167.4 |
| 2019 | 36.2 | 165.1 | 51.9 | 63.9 | 205.6 | 384.4 | 264.4 | 191.2 |
| 2020 | 47 | 222.4 | 100.1 | 98 | 228.7 | 417 | 265.2 | 223.1 |
| Average | 74.8 | 137.8 | 85.1 | 89.5 | 135.2 | 233.3 | 150.7 | 140.6 |

Source: The data in Table 3.

*3.2. Estimating the Productivity of Water and Energy Used in the Agricultural Sector during the Period 1995–2020*

The value of productivity per unit of water and energy at the level of the agricultural sector is estimated by dividing the value of agricultural output by the used quantities of water and energy during the period 1995–2020. It is clear from the data in Table 5 that the value of water productivity increased from 2568.3 riyals/thousand m$^3$ in 1995 to 7080.8 riyals/thousand m$^3$ in 2020; i.e., it increased at an annual growth rate of 7.03% during the study period. The value of diesel productivity also increased from 2608.8 thousand riyals/barrel in 1995 to 6958.0 thousand riyals/barrel in 2020; i.e., it increased at an annual growth rate of 6.67% during the study period. As for the value of electricity production, it decreased from 23.7 million riyals/gigawatt-hour in 1995 to 11.7 million riyals/gigawatt-hour in 2020; i.e., it decreased at an annual rate of 2.02% during the study period.

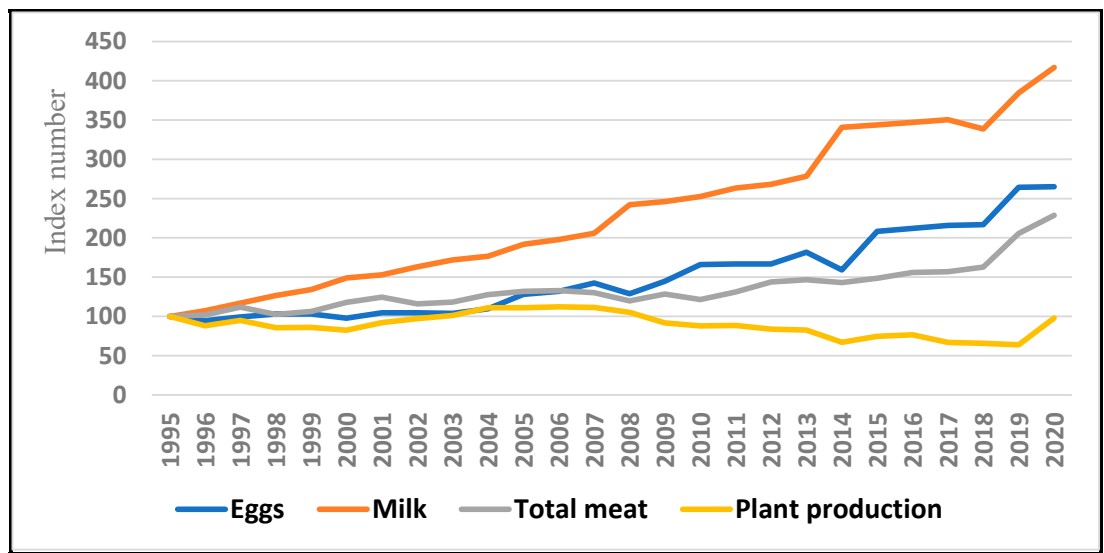

**Figure 2.** The index of vegetable and animal food production during the period 1995–2020. Source: The data in Table 4.

**Table 5.** Value of water and energy productivity used in the agricultural sector during the period 1995–2020.

| Year | Water (Riyals/Thousand m³) | Electricity (Million Riyals/GWh) | Diesel (Thousand Riyals/Barrel) |
|------|------|------|------|
| 1995 | 2568.3 | 23.7 | 2608.8 |
| 1996 | 2476.4 | 22.1 | 2887.3 |
| 1997 | 2094.9 | 22.9 | 2762.6 |
| 1998 | 2186.6 | 20.7 | 3117.5 |
| 1999 | 2205.8 | 18.8 | 2937.9 |
| 2000 | 2330.3 | 18.5 | 3344.9 |
| 2001 | 2263.0 | 17.7 | 3108.5 |
| 2002 | 2337.2 | 16.2 | 3116.3 |
| 2003 | 2388.9 | 16.2 | 3162.4 |
| 2004 | 2247.7 | 15.3 | 3398.0 |
| 2005 | 2425.4 | 14.4 | 3636.1 |
| 2006 | 2679.1 | 13.5 | 3785.9 |
| 2007 | 3011.1 | 13.8 | 3856.4 |
| 2008 | 3119.9 | 12.8 | 4324.3 |
| 2009 | 3222.6 | 8.9 | 5083.7 |
| 2010 | 3615.4 | 13.9 | 5763.1 |
| 2011 | 3416.7 | 13.8 | 6193.5 |
| 2012 | 3203.7 | 12.9 | 6718.1 |
| 2013 | 3108.2 | 13.5 | 7446.8 |
| 2014 | 3028.1 | 13.0 | 5062.4 |
| 2015 | 2868.2 | 11.6 | 5137.1 |
| 2016 | 3038.0 | 11.2 | 5228.0 |
| 2017 | 3147.0 | 10.7 | 5994.2 |
| 2018 | 3184.3 | 12.3 | 6211.6 |
| 2019 | 5828.8 | 12.3 | 6368.6 |
| 2020 | 7080.8 | 11.7 | 6958.0 |

Source: The data in Tables 1 and 2.

The productivity of the water and energy unit at the level of plant production was estimated by dividing the amount of plant food production by the quantities used of water and energy during the period 1995–2020. It is clear from the data in Table 6 that water productivity increased from 433.0 kg/thousand m$^3$ in 1995 to 740.0 kg/thousand m$^3$ in 2020; i.e., it increased at an annual growth rate of 2.84% during the study period. Diesel productivity also increased, from 439.8 tons/barrel in 1995 to 727.2 tons/barrel in 2020, i.e., an annual growth rate of 2.61% during the study period, while electricity productivity decreased, from 4.0 tons/megawatt-hour in 1995 to 1.22 tons/megawatt-hour in 2020; i.e., it decreased at an annual rate of 2.78% during the study period.

**Table 6.** Productivity of water and energy used in food production during the period 1995–2020.

| Year | Vegetarian Food | | | Red Meat | | |
|---|---|---|---|---|---|---|
| | Water (kg/thousand m$^3$) | Electricity (tons/MWh) | Diesel (tons/barrel) | Water (kg/thousand m$^3$) | Electricity (tons/MWh) | Diesel (tons/barrel) |
| 1995 | 433.0 | 4.00 | 439.8 | 10.39 | 0.10 | 10.56 |
| 1996 | 369.2 | 3.30 | 430.5 | 10.12 | 0.09 | 11.80 |
| 1997 | 326.4 | 3.57 | 430.5 | 8.41 | 0.09 | 11.10 |
| 1998 | 304.2 | 2.88 | 433.7 | 8.70 | 0.08 | 12.40 |
| 1999 | 301.5 | 2.57 | 401.5 | 8.69 | 0.07 | 11.57 |
| 2000 | 293.7 | 2.33 | 421.6 | 8.89 | 0.07 | 12.76 |
| 2001 | 317.1 | 2.48 | 435.6 | 8.58 | 0.07 | 11.79 |
| 2002 | 341.1 | 2.36 | 454.7 | 8.86 | 0.06 | 11.82 |
| 2003 | 360.3 | 2.44 | 477.0 | 9.15 | 0.06 | 12.11 |
| 2004 | 359.0 | 2.44 | 542.8 | 8.41 | 0.06 | 12.72 |
| 2005 | 383.5 | 2.27 | 574.9 | 9.09 | 0.05 | 13.63 |
| 2006 | 424.0 | 2.13 | 599.2 | 10.00 | 0.05 | 14.13 |
| 2007 | 463.4 | 2.12 | 593.4 | 11.09 | 0.05 | 14.20 |
| 2008 | 447.6 | 1.83 | 620.4 | 11.27 | 0.05 | 15.63 |
| 2009 | 399.3 | 1.11 | 629.8 | 11.59 | 0.03 | 18.29 |
| 2010 | 391.5 | 1.50 | 624.0 | 11.94 | 0.05 | 19.03 |
| 2011 | 355.4 | 1.44 | 644.2 | 10.71 | 0.04 | 19.41 |
| 2012 | 306.9 | 1.23 | 643.6 | 9.88 | 0.04 | 20.72 |
| 2013 | 284.3 | 1.24 | 681.2 | 9.33 | 0.04 | 22.37 |
| 2014 | 219.1 | 0.94 | 366.2 | 12.65 | 0.05 | 21.14 |
| 2015 | 230.2 | 0.93 | 412.4 | 12.39 | 0.05 | 22.18 |
| 2016 | 248.2 | 0.91 | 427.1 | 13.24 | 0.05 | 22.78 |
| 2017 | 223.6 | 0.76 | 425.9 | 13.91 | 0.05 | 26.49 |
| 2018 | 222.0 | 0.86 | 433.0 | 14.21 | 0.06 | 27.72 |
| 2019 | 390.7 | 0.82 | 426.9 | 26.19 | 0.06 | 28.62 |
| 2020 | 740.0 | 1.22 | 727.2 | 33.88 | 0.06 | 33.29 |

Source: The data in Tables 1–3.

The productivity of the water and energy unit at the level of red meat production was calculated by dividing the amount of produced red meat by the amount of consumed water and energy between 1995 and 2020. The same data in Table 6 show that water productivity grew from 10.39 kg/thousand m$^3$ in 1995 to 33.88 kg/thousand m$^3$ in 2020, representing a 9.04% annual growth rate during the research period. Diesel productivity

increased from 10.56 tons/barrel in 1995 to 33.29 tons/barrel in 2020, representing an annual growth rate of 8.61% over the study period, whereas electricity productivity decreased from 0.1 tons/megawatt-hour in 1995 to 0.06 tons/megawatt-hour in 2020, representing an annual growth rate of 0.06% over the study period.

*3.3. Measuring the Correlation Coefficients between Food Production and Water and Energy Consumption in the Agricultural Sector*

A simple correlation coefficient was calculated between the value of agricultural output and the food production index (plant and animal) and water and energy consumption in Saudi agriculture from 1995 to 2020 to determine the extent of economic interdependence between food production on one hand and water and energy consumption on the other. Table 7 shows that there is a negative correlation between the value of agricultural output and the index of food production and water and diesel consumption, while a positive correlation was discovered between the value of agricultural output and the index of food production and electricity consumption. The simple correlation coefficient assesses the degree and direction of a relationship between two variables in a given set of circumstances.

**Table 7.** Matrix of simple correlation coefficients between the value of agricultural output, the food production index, and the consumption of water, electricity, and diesel in Saudi agriculture during the period 1995–2020.

| Variable | Agricultural Production Value $Y_1$ | Diesel $X_3$ | Electricity $X_2$ | Water $X_1$ |
|---|---|---|---|---|
| Agricultural Production Value $Y_1$ | 1 | −0.14 | 0.93 | −0.77 |
| Water $X_1$ | −0.14 | 1 | −0.13 | 0.36 |
| Electricity $X_2$ | 0.93 | −0.13 | 1 | −0.77 |
| Diesel $X_3$ | −0.77 | 0.36 | −0.77 | 1 |
| Variable | Food Production Index $Y_2$ | Water $X_1$ | Electricity $X_2$ | Diesel $X_3$ |
| Food Production Index $Y_2$ | 1 | −0.38 | 0.89 | −0.75 |
| Water $X_1$ | −0.38 | 1 | −0.13 | 0.36 |
| Electricity $X_2$ | 0.89 | −0.13 | 1 | −0.77 |
| Diesel $X_3$ | −0.75 | 0.36 | −0.77 | 1 |

Source: The data in Tables 1, 2 and 4.

It is clear from the data in Table 8 that the partial correlation coefficient between the value of agricultural output and the amount of water used was 0.047, excluding the effects of both electricity and diesel. The partial correlation coefficient between the value of agricultural output and electricity consumption was 0.818, excluding the effect of water and diesel. The partial correlation coefficient between the value of agricultural output and diesel consumption was −0.228, excluding the effect of both water and electricity. It is also clear from the data in Table 8 that the partial correlation coefficient between the index of food production and the amount of water used was 0.555, excluding the effect of both electricity and diesel. The partial correlation coefficient between the index of food production and electricity consumption was 0.824, excluding the effect of water and diesel. The partial correlation coefficient between the index of food production and diesel

consumption was 0.025, excluding the effect of water and electricity. From the above, it is clear that about 55.5%, 82.4%, and 2.5% of the changes that occurred in the food production index in Saudi agriculture are attributed to changes in the consumption of water, electricity, and diesel, respectively.

**Table 8.** The partial correlation coefficient between the value of agricultural output, the food production index, and the consumption of water and energy in Saudi agriculture during the period 1995–2020.

| Food Production Index | Agricultural Production Value | First-Order Partial Correlation Coefficient |
|:---:|:---:|:---:|
| −0.585 | −0.052 | $r_{YX_1/X_2}$ |
| −0.222 | −0.230 | $r_{YX_3/X_2}$ |
| 0.411 | 0.411 | $r_{X_1X_3/X_2}$ |
| 0.917 | 0.929 | $r_{YX_2/X_1}$ |
| −0.711 | −0.779 | $r_{YX_3/X_1}$ |
| −0.782 | −0.782 | $r_{X_2X_3/X_1}$ |
| −0.711 | −0.779 | $r_{YX_3/X_1}$ |
| 0.917 | 0.929 | $r_{YX_2/X_1}$ |
| −0.782 | −0.782 | $r_{X_2X_3/X_1}$ |
| **Food Production Index** | **Agricultural Production Value** | **Second-Order Partial Correlation Coefficient** |
| 0.555 | 0.047 | $r_{YX_1/X_2X_3}$ |
| 0.824 | 0.818 | $r_{YX_2/X_1X_3}$ |
| −0.025 | −0.228 | $r_{YX_3/X_1X_2}$ |

Source: The data in Table 7.

### 3.4. Estimating the Proposed Model for the Economic Nexus between Food Production and Energy and Water Consumption

The suggested model was calculated by using the ordinary least squares (OLS) method from 1995 to 2020 to investigate the relationship between water and energy consumption on one hand and plant and animal food production on the other. The equations of the proposed model in Table 9 show the following: (1) a change of 10% in the cropped area (X1) results in a change of 3.23% in the amount of water used; (2) a change of 10% in the cumulative number of projects funded by the Agricultural Development Fund (X2) results in a change of 3.14% in the amount of electricity consumed; (3) a change of 10% in the value of fixed capital for machines and engines as an alternative variable for the number of machines and engines (X3) results in a change of 1.57% in diesel consumption; and (4) a change of 10% in the estimated consumption of water, electricity, and diesel results in a change of 1.97%, 2.78%, and 0.73% in the index of plant and animal food production, respectively. The equations of the proposed model are free from the problem of autocorrelation to the residuals, and they also have good efficiency in representing the data used in the estimation, according to the indicators of measuring the efficiency of the model, the most important of which is the inequality coefficient of Theil's U, whose value is close to zero (see Table 10).

**Table 9.** Statistical estimation of the equations of the proposed model to study the correlation between food production and water and energy consumption during the period 1995–2020.

| Endogenous Variables | Equation |
|---|---|
| Water | $\text{Ln}\hat{Y}_1 = 0.486 + 0.323\text{Ln}X_1 + 0.814\ AR(1)$ |
| | $(0.14)\ ^{ns}\ (2.79)\ ^{**}\ (4.51)\ ^{**}$ |
| | $R^2 = 0.53\ F = 8.15\ D.W = 1.26$ |
| Electricity | $\text{Ln}\hat{Y}_2 = 4.849 + 0.314\text{Ln}X_2 + 0.749\ AR(1)$ |
| | $(2.56)\ ^{**}\ (2.76)\ ^{**}\ (3.72)\ ^{**}$ |
| | $R^2 = 0.92\ F = 80.80\ D.W = 2.17$ |
| Deiseal | $\text{Ln}\hat{Y}_3 = 3.108 + 0.157\text{Ln}X_3 + 0.498\ AR(1)$ |
| | $(9.25)\ ^{**}\ (2.19)\ ^{*}\ (2.36)\ ^{*}$ |
| | $R^2 = 0.72\ F = 18.71\ D.W = 1.78$ |
| Food Production Index | $\text{Ln}\hat{Y}_4 = -6.978 + 0.197\text{Ln}\hat{Y}_1 + 0.278\text{Ln}\hat{Y}_2 + 0.073\text{Ln}\hat{Y}_3 + 0.729\ AR(1)$ |
| | $(-1.98)\ ^{*}\ (2.03)\ ^{*}\ (2.71)\ ^{**}\ (2.12)\ ^{*}\ (3.30)\ ^{**}$ |
| | $R^2 = 0.94\ F = 50.45\ D.W = 1.71$ |

** = significant at 1% probability level, * = significant at 5% probability level, ns = not significant. Source: Statistical analysis of the data in this study.

**Table 10.** Efficiency indicators of the proposed model to study the correlation between food production and water and energy consumption.

| Index | First | Second | Third | Fourth |
|---|---|---|---|---|
| Root-Mean-Square Error (RMSE) | 0.221 | 0.186 | 0.114 | 0.065 |
| Mean Absolute Error (MAE) | 0.191 | 0.165 | 0.093 | 0.050 |
| Mean Absolute Percentage Error (MAPE) | 6.884 | 2.044 | 3.988 | 0.998 |
| Coefficient of Uncertainty (Theil's U) | 0.040 | 0.011 | 0.023 | 0.006 |

Source: It was collected and calculated from the equations of the proposed model in Table 9.

## 4. Conclusions

By studying the current situation, it was found that the index of plant food production declined from 100% in 1995 to 63.9% in 2019. This was due to the decisions issued on the rationalization of water consumption in Saudi agriculture and the restructuring of the crop structure at the level of regions and governorates within each administrative region. As for animal production, the results showed an increase in the index of red meat, poultry meat, milk, and eggs. In general, food production (vegetable and animal) increased in 2020 compared with its counterpart in 1995, at a rate of 123.1%, i.e., an annual growth rate of 4.92% during the period 1995–2020.

By calculating the partial correlation coefficient of the second order between food production and water and energy consumption during the period 1995–2020, it was found that about 55.5%, 82.4%, and 2.5% of the changes that occurred in the index of plant and animal food production were attributed to changes in the consumption of water, electricity, and diesel, respectively. By estimating the proposed model to study the correlation between water and energy consumption on one hand and the index of plant and animal food production on the other during the period 1995–2020, it was found that an expansion in the consumption of water, electricity, and diesel by 10% led to an increase in the index of food production by 1.97%, 2.78%, and 0.73%, respectively.

In view of the scarcity of water resources and the issuance of decisions to rationalize the use of water in Saudi agriculture, the rationalization of water consumption is expected to continue, so that its consumption does not exceed the amount of renewable groundwater of 8 billion m$^3$. On 27 March 2021, His Royal Highness Crown Prince Mohammed bin

Salman announced the Green Middle East Initiative. The initiative included reducing carbon emissions by 278 million tons by 2030 and increasing the use of renewable energy in various economic sectors. In light of the Green Middle East Initiative, the quantities of diesel used are expected to reduce and the consumption of electricity in the agricultural sector is expected to expand. There is no doubt that rationalizing water consumption and reducing diesel consumption affects electricity consumption and the production of plant and animal food.

Thanks to the results of this study, it can be said that the interdependence between water, energy, and food has become relevant to the environmental problems that the Kingdom of Saudi Arabia suffers from, in particular the problem of water scarcity and the trend toward reducing carbon emissions through the implementation of the Middle East Green Initiative. In light of the strong interdependence between water, energy, and food production, the agricultural policy has become necessary to increase the amount supplied or available to be used in food production, in addition to expanding the production of clean energy and its use in the agricultural sector.

**Author Contributions:** Conceptualization, K.A. and Y.A.; methodology, A.G.; software, S.K.; validation, S.A. and N.A.; formal analysis, Y.A.; investigation, A.G.; data curation, S.A.; writing—original draft preparation, S.A. and N.A.; writing—review and editing, K.A. and Y.A.; supervision, A.G.; software, S.K. All authors have read and agreed to the published version of the manuscript.

**Funding:** This research received no external funding.

**Data Availability Statement:** No new data were created or analyzed in this study. Data sharing is not applicable to this article.

**Acknowledgments:** This research was supported by the King Saud University Deanship of Scientific Research, College of Food and Agricultural Sciences Research Centre. The authors thank the Deanship of Scientific Research and RSSU at King Saud University for their technical support.

**Conflicts of Interest:** The authors declare no conflict of interest.

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
