# Peer review of "The Economic Nexus between Energy, Water Consumption, and Food Production in the Kingdom of Saudi Arabia"

_economies, doi:10.3390/economies11040113_

Round 1

Reviewer 1 Report

REVIEWER 1. “The Economic Nexus Between Energy, Water Consumption and Food Production in The Kingdom of Saudi Arabia”.

Summary

This is an empirical paper that estimates the relationship between energy, water use, and plant and animal food production in Saudi Arabia from 1995 to 2020. The autor uses econometric analysis and the partial correlation coefficient to investigate wether changes in the index of plant and animal food production is correlated with changes in the consumption of water, electricity and diesel. This study finds that about 55.5%, 82.4%, and 2.5% of changes in the index of plant and animal food production were related to changes in the consumption of water, electricity, and diesel, respectively.

Comments

 1) The main objection to this paper is that it does not clarify his novel contribution with respect to previous research. I am also not convinced by the empirical analysises. It uses data too aggregated and without prior cleaning. For example, it would be desirable to disaggregate plant production (cereals, fruits and vegetables) since water consumption is very different depending on the type of plant production and this affects the productivity of different crops. On the other hand, since the price indices of the different productions (vegetables and animals) are known, the production value could be deflated and productivity measures obtained at constant prices.

2) You have to deflate the value of production to carry out a robust analysis of productivity.

3) Likewise, in my opinion, the food production index (vegetables and animals) used, is not adequate and produces biased results. In the same sense, the productivity measures presented are not appropriate.

4) Another aspect to consider is whether the existence of a high correlation between the variables of the model would lead to biased estimators.

Research should address these issues.

5) There are some aspects of structure that need to be modified in order to differentiate between the purely metodological aspects and the results of the present investigation. For example, methodological aspects related to the calculation of simple and partial correlation coefficients or the estimation of an econometric model are presented as objectives of the work. In relation to the model, I think it is necessary to differentiate the consumption of water in plant production and animal production.

6) It does not previously define the variables used to calculate the correlation coefficients.

7) The author must explain the meaning of the links obtained between the variables. About the presentation of results, the author should comment he coefficients and their meanings together in order to be much more comprehensive.

8) A section of conclusions should be introduced.

9) In figures, commas must be used to separate thousands.

 In light of the comments, I do not recommend this article for publication in its current state. Previously, the above aspects must be incorporated.

Author Response

Respond to the reviewers

The research team extends its sincere thanks and appreciation to the editorial board of the esteemed journal and to the reviewers, and we thank them for the effort made in reviewing the research. The research team would like to clarify the following points:

First reviewer:

  1. The reviewer believes that the research paper does not explain its new contribution to previous studies. This was clarified by adding a paragraph at the end of the introduction. The reviewer also believes that he is not convinced of the analyzes based on the collected data, and this opinion belongs to the reviewer only, because all economic research in all countries of the world relies on published data in estimating models. Dividing vegetable production into grains, fruits, and vegetables under the pretext that water consumption is different. This is true in the case of comparison between grains, fruits, and vegetables when water consumption is low in the agricultural sector. This point is outside the scope and objectives of the research. Also, the reviewer believes that the index numbers are known, and this is correct, and we would like to point out that the index numbers are the most appropriate and best indicator for studying changes in a phenomenon, whether prices or quantities. This methodology was used in calculating the food production index.
  2. The reviewer believes that the value of production must be reduced to perform a robust analysis, and this is incorrect, because the use of current prices or fixed prices does not affect the nature of the results. The use of current prices reflects the current situation.
  3. The reviewer believes that the food production index is insufficient and results in biased results, which is incorrect, as the food production index considers all vegetable production (grains, fruits and vegetables), red and white meat (poultry and fish), eggs and milk. And through everything for plant and animal production, included in tables (3, 4).
  4. The reviewer believes that a significant correlation between the variables affects the estimates. And the research team asks where is the big correlation? This problem does not exist at all, see the correlation coefficients mentioned in table (8), especially between the variables of water, electricity and diesel, as they are all less than 0.8 and therefore their use does not affect the estimates because they are internal variables in the proposed model.
  5. The reviewer considers presenting correlation coefficients or the proposed model as objectives of the research paper. The research team believes that it is necessary to set them as sub-goals as stated in the body of the research, because this study relies on econometric analysis that combines statistics, math and economics, to increase the depth of analysis and measurement and the extent of the comprehensiveness of internal and external variables. As for the extent to which water used in plant production is separated from that used in animal production. The research team confirms that the water used in plant production cannot be separated from animal production, and no country in the world has published data on the water used in plant production and that used in animal production.
  6. The reviewer believes that the variables used in calculating correlation coefficients were not pre-determined, and this is incorrect, because table (7) explains the variables in terms and symbols.
  7. The reviewer considers that the meaning of the correlation coefficients obtained must be explained. The research team believes that the explanation of the meaning of correlation coefficients was already in included in the research, specifically on before Table (8).
  8. Conclusion is included in the research.
  9. Numbers in graphs (1,2) do not require commas to be placed between the numbers.

Reviewer 2 Report

The authors must expand the Literature review by making reference to the results obtained by others, in different countries, on similar topic:  the nexus between energy, water consumption and food production. If the relevant international literature does not exist, the authors must emphasize this and underline the novelty of their study.

The authors must add a subsection of conclusions to emphasize the most important results and their practical and scientific relevance.

The bibliography must be completed accordingly, with the international literature.

Author Response

Respond to the reviewers

The research team extends its sincere thanks and appreciation to the editorial board of the esteemed journal and to the reviewers, and we thank them for the effort made in reviewing the research. The research team would like to clarify the following points:

Second reviewer:

  1. Three recent studies have been added from different countries.
  2. A paragraph has been added explaining the importance of the practical and scientific results of the research paper.
  3. Conclusion added in the research.

Round 2

Reviewer 1 Report

Comments to the Author

The authors have not fully addressed comments and suggestions of my previous report, so the revised paper has not been sufficiently improved. Only, three new references has been included in the introduction, but they are not adequately connected with the rest of the introduction.

Specifically, I think this work there is not a novel contribution to previous research. In addition, the revised version has not addressed the recommendations to improve the empirical analysis. At this point I refer to the comments of the previous review:

1) It uses data too aggregated and without prior cleaning. For example, it would be desirable to disaggregate plant production (cereals, fruits and vegetables) since water consumption is very different depending on the type of plant production and this affects the productivity of different crops. On the other hand, since the price indices of the different productions (vegetables and animals) are known, the production value could be deflated and productivity measures obtained at constant prices.

2) You have to deflate the value of production to carry out a robust analysis of productivity.

3) Likewise, in my opinion, the food production index (vegetables and animals) used, is not adequate and produces biased results. In the same sense, the productivity measures presented are not appropriate.

4) Another aspect to consider is whether the existence of a high correlation between the variables of the model would lead to biased estimators.

5) In relation to the model, I think it is necessary to differentiate the consumption of water in plant production and animal production.

6) The authors must explain the meaning of the links obtained between the variables. About the presentation of results, the author should comment he coefficients and their meanings together in order to be much more comprehensive.

8) I think the authors should put more effort into the conclusions. Beyond a summary of the article, they should include recommendations based on their findings. For example, specific agricultural policy actions against the impact of a rise in energy costs.

9) In figures, commas must be used to separate thousands.

Author Response

Respond to the comments of the first reviwer

The research team extends its sincere thanks and appreciation to the editorial board of the esteemed journal and to the reviewers, and we thank them for the effort exerted in reviewing and arbitrating the research. The research team would like to clarify the following points:

First reviewer:

  • Three previous studies have been added, by adding a paragraph at the end of the introduction. And all economic research in all countries of the world depends on published data in estimating models. The food plant production was also divided into grains, fruits and vegetables - Table (3).
  • The value of production at current prices has been changed to the value at constant prices (2010 = 100), Table (2). The index for the production of cerals, fruits and vegetables has also been added in Table (4). The value of water, electricity and diesel productivity has also been recalculated in Table (5) and the accompanying explanation in the body of the research according to fixed prices.
  • The food production index considers vegetable production (cereals, fruits and vegetables), red and white meat (poultry and fish), eggs and milk.
  • The reviewer believes that the presence of a significant correlation between the variables affects the estimates. See the correlation coefficients presented in Table (8), especially between the variables of water, electricity and diesel, as they are all less than 0.8 and therefore their use does not affect the estimates because they are internal variables in the proposed model.
  • Regarding the extent to which water used in plant production is separated from that used in animal production. The research team confirms that the water used in plant production cannot be separated from animal production, and no country in the world has published data on the water used in plant production and that used in animal production.
  • The variables used in calculating the correlation coefficients have been pre-determined, and Table (7) shows the variables in terms and symbols.
  • An explanation of the meaning of correlation coefficients is already in the body of the research, specifically before Table (8).
  • Conclusion is added in manuscript.
  • Numbers in graphs (1,2) do not require commas to be placed between the numbers.

Reviewer 2 Report

The authors must expand the Literature review by making reference to the results obtained by others, in different countries, on similar topic:  the nexus between energy, water consumption and food production. This conclusion is not supported by studies: ”By reviewing the methodology and results of previous studies, it was found that some studies relied on the calculation of simple correlation coefficients, while others used the one-equation model. This study was distinguished from all previous studies, in that it used partial correlation coefficients of the first and second order, and it also used a proposed model consisting of four behavioral equations, which include internal and external variables, in order to be more comprehensive to study the interdependence between water, energy and food production.”

This study is not relevant for the current article: ”Saul Ngarava (2021). Long term relationship between food, energy and water inflation in South Africa, Water- Energy 398 Nexus 4,”

Studies like those conducted by World Band (The Water-Energy-Food Nexus in the Middle East and North Africa)2018; Food and Agriculture Organization of The United Nations (2014) should be a starting point.

The bibliography must be completed accordingly, with the relevant literature.

Author Response

Response to second reviewer

  • Three studies have been added from different countries.
  • A paragraph has been added explaining the importance of the practical and scientific results of the research paper

Round 3

Reviewer 1 Report

Comments

- I suggest that the authors improve the scientific justification of the paper in the introduction section, incorporating the following papers:

FAO (2014). The Water-Energy-Food Nexus. A new approach in support of food security and sustainable agriculture.

Mahlknecht, J.; González-Bravo, R. and Loge, F. (2020). Water-energy-food security: A Nexus perspective of the current situation in Latin America and the Caribbean. Energy, 194.

- References also need to be checked and alphabetized.

Author Response

Response to reviwer#1

Thank you for sending us two papers related to our manuscript for improving it.

  • We add two suggested papers added it in the introduction.
  • Also, we check the references and alphabetized them accordingly.

Kind regards

Reviewer 2 Report

Good for publishing

Author Response

The reviwer check all items improved